# Role of Prenatal Nutrition in the Development of Insulin Resistance in Children

**DOI:** 10.3390/nu15010087

**Published:** 2022-12-24

**Authors:** Annalisa Blasetti, Alessia Quarta, Miriana Guarino, Ilenia Cicolini, Daniela Iannucci, Cosimo Giannini, Francesco Chiarelli

**Affiliations:** Department of Paediatrics, University of Chieti—Pescara, G. d’Annunzio, 66100 Chieti, Italy

**Keywords:** prenatal nutrition, insulin resistance, micronutrients

## Abstract

Nutrition during the prenatal period is crucial for the development of insulin resistance (IR) and its consequences in children. The relationship between intrauterine environment, fetal nutrition and the onset of IR, type 2 diabetes (T2D), obesity and metabolic syndrome later in life has been confirmed in many studies. The intake of carbohydrates, protein, fat and micronutrients during pregnancy seems to damage fetal metabolism programming; indeed, epigenetic mechanisms change glucose−insulin metabolism. Intrauterine growth restriction (IUGR) induced by unbalanced nutrient intake during prenatal life cause fetal adipose tissue and pancreatic beta-cell dysfunction. In this review we have summarized and discussed the role of maternal nutrition in preventing insulin resistance in youth.

## 1. Introduction

### 1.1. Definition

A Consensus Conference has defined insulin resistance (IR) as “the decreased tissue response to insulin-mediated cellular actions, so it is the inverse of insulin sensitivity” [1]. Therefore, IR represents a state in which normal plasma insulin concentrations are not able to maintain peripheral glucose removal and inhibit both hepatic glucose output and hepatic glucose-induced output of very low-density lipoproteins [1,2]. The impaired glucose metabolism in the liver and the declining beta-cell function play a crucial role in driving type 2 diabetes (T2D) development [3,4]. Thus, although IR represents the first step in the natural history of the spectrum of impaired glucose metabolism, so far there are no standard values accepted worldwide for IR in children; also, its frequency varies according to gender and race [5]. However, surrogate markers are widely adopted in the clinical setting, offering relevant tools that are useful for characterizing the risk of developing a complex spectrum of metabolic alteration.

### 1.2. Measurement

The evaluation of peripheral insulin sensitivity in vivo consists of the application of direct methods, such as the hyperinsulinemic-euglycemic clamp, the pancreatic suppression test and the minimal model approximation of the metabolism of glucose (MMAMG) [6]. The hyperinsulinemic-euglycemic clamp study is the “gold-standard” for whole body insulin sensitivity assessment; this method measures insulin-stimulated glucose disposal at a specific concentration of hyperinsulinemia [7]. Unfortunately, direct methods are elaborate, invasive, time-consuming and costly, so they are only used for research purposes [8].On the other hand, fasting plasma insulin (FPI), fasting insulin resistance index (FIRI), McAuley index (McA), homeostasis model assessment for insulin resistance (HOMA-IR), quantitative insulin-sensitivity check index (QUICKI), insulin sensitivity index (ISI) and whole body insulin sensitivity index (WBISI) are indirect methods that measure plasma insulin levels during fasting or after oral glucose load (Table 1) [9]. HOMA-IR is calculated as the result of the fasting insulin−glucose product divided by a different constant based on the measurement unit; it is the simplest method, and it depends on both the fasting glucose and insulin concentration [7]. All of these indexes have been validated according to the gold standard techniques, are well correlated with glucose-related metabolic abnormalities and might be used in clinical settings.

### 1.3. Pathophysiology

Insulin induces metabolic effects on target cells via PI3K (phosphoinositide-3-kinase)-Akt (protein kinase B) signaling, influencing glycogen synthesis, glycolysis, fatty acid and protein synthesis [10]. PI3K-Akt genes and many other genes might be involved in the pathophysiology of IRlike PPARG (peroxisome proliferator-activated receptor-γ), IRS1 (insulin receptor substrate 1) and GCKR (glucokinase regulator, [11] but at the present time data are still controversial [12]). The most important consequence of IR is the presence of a compensatory hyperinsulinemia state, characterized by an excessive and delayed rise in insulin secretion after meals, leading to triglyceride accumulation in hepatic and muscle tissues and a reduction of GLUT-4 translocation [13]. The link between an unfavorable prenatal environment and adverse health outcomes later in life has been confirmed in many studies. Indeed, poor nutritional status during the fetal period may cause structural and functional alterations in many organs, such as the liver, brain, muscle, pancreas and adipose tissue [14]. This hypothesis is known as “predictive adaptive response” (PAR) and describes the process by which the prenatal environment models fetal development and leads the organism to express a metabolic spearing phenotype based on the expected future environmental conditions [15]. Intrauterine growth restriction (IUGR) is the result of impaired fetal growth due to poor or unbalanced nutrient intake during intrauterine development [16]. This condition has many metabolic consequences regarding insulin activity; indeed, peripheral insulin sensitivity decreases, insulin-stimulated protein synthesis in muscles reduces and hepatic glucose production increases. If these conditions are persistent, the nutrient uptake and storage may lead to several complications later in life, such as obesity, T2D and IR [16,17]. These considerations are the core of the “thrifty phenotype” hypothesis, proposed by Hales and Barker in 1992 [18].

### 1.4. Maternal Nutrition and Intrauterine Fetal Growth

Maternal nutrition during pregnancy plays a fundamental role in both providing the fetus the essential nutrients for growth and establishing the phenotypic and metabolic characteristics of the fetus during postnatal life (Figure 1) [19]. The correlation between nutrition during different ages of life and the probability of developing metabolic alterations such as IR, obesity, hypertension and T2D, is now well known. Intrauterine nutrition represents the first phase and probably one of the most important and decisive ones for the metabolic programming of the fetus [20]. Fetal exposure to an inadequate or unbalanced intake of macro and micronutrients might cause changes in the fetal epigenome. Epigenetic changes determine modifications in the gene expression and, consequently, in the metabolic programming of the fetus, predisposing it to the onset of metabolic diseases later in life. It has been observed that the early stages of embryo development represent a time window in which cells are most susceptible to epigenetic changes [21]. However, despite the fact that the first trimester represents a crucial period in the metabolic programming of the fetus, it is also true that fetal exposure to unfavorable intra-uterine conditions continues to influence epigenetic modifications in the next stages of development. Indeed, it has been shown that inadequate intake of micronutrients like vitamin B12 and B9 during the second trimester of pregnancy are associated with an increased HOMA-IR in the offspring. According to the thrifty phenotype theory, intrauterine malnutrition could be correlated with an increased risk of obesity onset and metabolic syndrome in postnatal life [22,23]. The result is an increased incidence of IR and T2D among younger subjects, leading to a raised cardiovascular risk and increased likelihood of metabolic complications in adulthood [24,25]. We know that those born small for gestational age (SGA) exhibit catch-up growth 6 months after birth, mainly via fat storage in the abdominal area. Moreover, these children demonstrated hyperinsulinemia and insulin resistance that predispose them to several metabolic diseases like T2D later in life. This condition may be explained by the thrifty phenotype hypothesis. Probably, this process is derived from DNA methylation, leading to the repression of gene expression involved in the regulation of glucose metabolism [26]. Furthermore, postnatal exposure to overnutrition contributes to the impairment of beta-cell function and insulin sensitivity. Many studies have demonstrated that maternal obesity represents an independent risk factor for the high BMI of offspring. Moreover, in humans, it has been demonstrated that infants exposed to overnutrition in utero could manifest an increased risks of obesity, diabetes and other complications, including non-alcoholic fatty liver disease (NAFLD) [27,28]. To better understand the concept of fetal malnutrition, we will discuss in more detail about some models of maternal diet to understand how these are correlated with the onset of metabolic alterations during postnatal life.

### 1.5. Maternal High Glycemix Index Diet

Numerous studies have demonstrated the correlation between maternal diet with a high glycemic content and the increased risk of developing IR in offspring [29]. Glycemic index (GI) represents the glycemic response after the ingestion of a carbohydrate-containing food [30,31]. Carbohydrates are distinguished in those that are absorbed quickly and cause a rapid rise in blood sugar (high GI) and others that release glucose more slowly (low GI). Sweets and processed grains are considered high-GI foods because they produce elevated blood glucose concentration, while foods rich in soluble fiber contribute to a lower GI. In adult populations, the intake of high GI and glycemic load (GL) foods has been associated with an increased risk of obesity and T2D [32]. Conversely, low-GI foods, such as fiber-containing foods, have been found to be protective [33,34]. It was seen that maternal blood glucose levels in pregnancy are more negatively influenced by the intake of high-GI carbohydrates. For this reason, primary interventions during pregnancy, such as dietary interventions, are important to improve perinatal outcomes. For example, using low-GI foods can reduce postprandial glycemia. Maslova et al. examined the association of maternal GI and GL during pregnancy with offspring body mass index (BMI) in the first 7 years of life in a large cohort study. It was observed that higher maternal GI and GL were likely related to an increase in IR and adiposity in children between 9 and 16 years exposed to maternal hyperglycemia in utero [35]. Moreover, it was supposed that a maternal diet with a high GI may influence the expression of leptin (LEP) and fat mass and the obesity-associated gene (FTO), as well as other appetite-related genes, like agouti-related peptide (*Agrp*), neuropeptide Y (*Npy*), pro-opiomelanocortin cocaine (*Pomc*), amphetamine-regulated transcript (*Cart*) and leptin receptor (*Lepr*) in specific tissues [36]. Leptin is a protein hormone synthesized in the white adipose tissue that acts on the hypothalamic receptors by regulating the sense of satiety [37,38,39]. Sideratou et al. observed using a mouse model that the glycemic qualities of carbohydrates consumed during pregnancy and lactation modifies the expression of appetite-related genes in the offspring. It was observed that an increased FTO expression is associated with a high GI diet during both pregnancy and postnatal ages; in particular, the expression is almost 4-fold higher in the placenta of female mice during pregnancy and >2-fold higher in the hypothalamus of pups fed with a high GI diet from weaning to about 2 years of age. By contrast, *Lep* expression in the visceral adipose tissue of offspring was >3-fold higher if mothers were fed with a low-GI diet throughout pregnancy and lactation [36]. These findings imply that consumption of rapidly digestible starchy foods may increase the risk of early-onset obesity and IR in humans. On the other hand, newer studies have proposed that a low-carbohydrate diet with an elevated content of fibers could be more effective for the prevention of T2D onset compared to a low-fat diet. Toh et al., in their study on mouse models, showed that a maternal high-fiber intake reduced the probability of diabetes onset in offspring that consumed a diabetogenic diet after weaning, especially in the male sex [40].

### 1.6. Maternal High-Fat Diet

Many animal studies have demonstrated the metabolic effects of a high-fat (HF) maternal diet on the fetus [41,42]. Saullo et al. have observed that offspring adiposity is significantly affected by a maternal HF diet and that the consumption of an HF diet during pregnancy influences the development of visceral white adipose tissue in offspring, especially when the pups also have an HF diet consumption during postnatal life [43,44]. Peng et al., in their study conducted on mice, demonstrated a correlation between fetal exposure to overnutrition and an increased risk for obesity and related metabolic diseases, such as hepatic steatosis and NAFLD [45]. In summary, a maternal HF diet leads to disrupted one-carbon (1C) metabolism in their children through epigenetic shifts, leading to metabolic gene expression changes. Methionine cycle dysfunction causes abnormal methyl group delivery to its substrate, modifying lipid metabolism through the turning off of the fatty acid oxidation process involving L-carnitine depletion and, thus, increasing the risk for offspring to develop NAFLD later in life. Moreover, a greater susceptibility of the male sex to disrupted 1C metabolism and methionine cycles has also been demonstrated [46]. Maternal overnutrition increases susceptibility to obesity and diabetes in their offspring [47]. In their study, Zhang et al. showed that maternal HF diet impairs placental fatty acid beta-oxidation (FAO) [48]. The placenta plays a central role in linking the mother and fetus, and it has been demonstrated that placental fatty acid metabolism influences the metabolic outcomes of the fetus [49]. It has been observed that maternal HF diet downregulates mRNA and protein expressions of carnitine palmitoyltransferase 2 (CPT2), an enzyme with an important role in FAO, by suppressing the AMPK/Sirt1/PGC1α signaling pathway in the placenta; therefore, decreased FAO in the placenta was related to an increased placental weight and a fetal growth restriction due to an intracellular lipid accumulation [50,51,52,53]. This would cause an increased risk of many metabolic changes in children at weaning age such as high body weight, glucose intolerance, hyperinsulinemia and hypercholesteremia.

### 1.7. Maternal Low-Protein Diet

The association between low-protein maternal diet and increased susceptibility to developing T2D during postnatal life has been noted in numerous human and animal studies [54]. Indeed, it has been observed that intrauterine exposure to malnutrition or inadequate nutrition is more associated with the development of reduced glucose tolerance in adult life. These observations clearly show that T2D can develop even in the absence of obesity [55,56]. In Western countries, the rise in popularity of vegan and vegetarian diets, which are characterized by a low-protein content, contributes to an increasing prevalence of maternal low-protein diets. Vegetarian mothers consume a low-protein diet [23,57] and give birth to babies with lower birth weight, exposing them to an increased risk of impaired glucose metabolism and T2D [58]. The correlation between fetal exposure to a reduced availability of nutrients and the development of impaired glucose metabolism in adulthood could be explained by the theory of the “thrifty phenotype” [59]. The dominant hypothesis in the metabolic programming of postnatal life attributes a central role to the fetal epigenome. In animal studies, it has been observed that intra-uterine exposure to a low-protein maternal diet determines modifications in methylations in the promoter regions of genes involved in glucose metabolism, influencing their expression, [60]. In studies conducted in mice exposed to low-protein diets during pregnancy, a reduction in pancreatic beta-cells and a consequent reduction in insulin synthesis in the offspring was observed. Indeed, the adequate function of pancreatic beta-cells depends both on their structural and functional integrity and a balanced nutritional rapport [61]. Consequently, the lack of protein in the maternal diet significantly contributes to the reduced secretion of insulin [62]. The reduced islet area and number of beta-cells are mainly due to the diminished expression of the FoxO1 and Pdx1 genes, or to the altered expression of the Reg1 pathway genes [63,64]. Furthermore, a low-protein maternal diet has demonstrated higher rates of beta-cell apoptosis [65,66,67] caused by increased oxidative stress and mitochondrial dysfunction [68]. Zambrano et al. have evidenced that a maternal low-protein diet was correlated to IR mainly in male rats; instead, the female rats had a greater insulin sensitivity [69]. This study hypothesized that the mechanism underlying the occurrence of IR in male rats could be mitochondrial dysfunction of pancreatic beta-cells [70]. These findings have also been observed in numerous human studies. Indeed, a higher incidence of T2D was observed in males, demonstrating that females are characterized by a greater insulin sensitivity and lower susceptibility to the onset of IR. These differences might depend on the protective effect of estrogens and this data could be supported by the fact that after menopause, the risk of T2D onset is comparable to that of men. Furthermore, sex chromosomes control adiposity and glucose homeostasis. The mechanisms by which estrogens rule insulin sensitivity are well known and consist of modifications of the signal transduction and metabolic pathways in both the central nervous system and fat, muscle and liver tissue [71]. A low-protein maternal diet not only has consequences for glucose metabolism, but also seems to affect the adequate development of skeletal muscle tissue and normal differentiation of bone marrow cells [72]. Furthermore, thyroid hormone synthesis and the hypothalamic−pituitary−gonadal axis are also influenced by a low-protein maternal diet [73]. The correlation between maternal malnutrition and thyroid dysfunction in offspring in later life has been highlighted in numerous studies. Indeed, it has been observed that a low-protein maternal diet, especially during lactation, is associated with the onset of hyperthyroidism; this could be due to an increased activity of 5-iodothyronine deiodinase, resulting in higher levels of triiodothyronine, the bioactive hormone [74]. Alterations in the reproductive function of offspring may be correlated to the impaired expression of genes associated with steroidogenesis, folliculogesis and steroid hormone receptors in the gonads [75,76]. The data in the literature show that a low-protein maternal diet leads to a decrease in the availability of essential amino acid levels in the maternal circulation and, consequently, in the fetus. Under these conditions, the fetus modifies its growth and metabolic programming. Therefore, protein intake is able to affect metabolic risk during childhood and adulthood through different pathways. Further studies are needed to completely characterize these metabolic alterations.

### 1.8. Role of Micronutrients

Micronutrients are nutritive substances that must be ingested in small quantities in the diet, and they are necessary for a series of physiological functions essential for the metabolism and biochemical processes of the cells [77]. An unbalanced diet may cause a reduced intake of micronutrients, resulting in negative effects on fetal growth and development (IUGR, low birth weight (LBW), congenital malformations, impaired cognitive development, immunodeficiencies) and maternal health [78,79]. Micronutrients are divided into vitamins and minerals. Impaired dietary intake or lack of any of them might lead to important consequences. Several clinical trials and perspective studies have examined the correlation of their unbalanced uptake with the risk of developing metabolic alterations during postnatal life. Particularly, a relevant inquiry has focused on the role of vitamin B12, folic acid and the levels of circulating homocysteine [80]. All of these elements play an essential role in the 1C metabolism [81]. Alterations in 1C metabolism seem to be implicated in epigenetic modifications and, consequently, in long-term metabolic programming [82]. One-carbon metabolism is characterized by interdependent metabolic pathways, such as the folate and methionine cycles, which regulate several cell cycle mechanisms, such as the synthesis of purine and pyrimidine bases and amino acids and the methylation of DNA, RNA and histones. The main mechanism underlying these metabolic pathways is represented by the transfer of a carbon unit (methyl group), and S-adenosyl methionine (SAM) represents the main coenzyme involved in methylation processes [83,84]. The one-carbon metabolism strictly depends on elements introduced with the diet, such as vitamins of group B (vitamin B9, B12, B6), methionine and betaine, each performing the function of cofactors or substrates. Vitamin B12 and vitamin B9 represent the major coenzymes for the methionine cycle, whereby homocysteine is converted to methionine by the enzyme methionine synthase following donation of the methyl group from 5-methyltetrahydrofolate, which is converted to tetrahydrofolate. In the case of vitamin B12 deficiency, 5-methyltetrahydrofolate cannot be converted to tetrahydrofolate and methionine synthesis cannot take place [85,86,87,88]. Vitamin B9 belongs to the group known as folates, a set of essential nutrients that participate in the synthesis of DNA and proteins. A deficiency of folic acid (and also of vitamin B6 and B12) induces an increase in homocysteine concentrations in the blood, a non-essential amino acid. Of note, several studies have clearly shown that high homocysteine concentrations result in neurotoxic, vasculotoxic and, therefore, teratogenic effects [89]. Vitamin B12 (also named cobalamin) is a water-soluble vitamin essential for red blood formation, homocysteine metabolism and the normal development and function of the central nervous system. Vitamin B12 and folate act as methyl donors in 1C metabolism, which influence cell growth and differentiation by DNA synthesis and epigenetic regulation [90]. Vitamin B12 and vitamin B9 deficiency is related to an increase in homocysteine levels. Another indicator of vitamin B12 deficiency is methylmalonic acid (MMA). Low vitamin B12 levels and high homocysteine levels during pregnancy are associated with an increased risk for LBW, SGA and IUGR and cardiovascular and metabolic disorders in adulthood (T2D, hypertension, coronary artery disease, etc.). Interestingly, the associations were time-dependent, showing a stronger correlation during the second trimester [91,92]. Moreover, vitamin B12 deficiency has been associated with congenital malformations such as neural tube defects, poor neurocognitive development and even maternal obesity and dyslipidemia. In addition, low maternal vitamin B12 level has been associated with lower offspring B12 concentrations in cord blood and during childhood as well as an increased risk of diabetes (IR in childhood) [93]. Indeed, an altered ratio of folate to vitamin B12 (high folate and low vitamin B12) has been associated with a high risk of gestational diabetes (GDM) with a consequent increased risk for the offspring of obesity and IR [92]. Finally, hyperhomocysteinemia could have an adverse effect on endothelial function mediated by oxidative stress. This can cause alterations in placental flow and, consequently, in fetal growth. Maher et al. have highlighted the relationship between folate and B12 status and GDM, which represents an alteration in glucose metabolism characterized by a state of IR and impaired glucose tolerance, diagnosed for the first time during the pregnancy [94,95]. The precise mechanisms linking the altered ratio folate/vitamin B12 and increased GDM risk is still not well known [96]. Vitamin B12 deficiency and blockage of methionine synthesis resulting in increased homocysteine levels may be contributing factors. A study into non-diabetic obese male and female adults found that B12 concentrations negatively correlated with fasting plasma glucose levels and the risk of IR [95,96]. Moreover, it was observed that high maternal folate concentrations at 28 weeks of gestation and low maternal B12 concentrations at 18 weeks of gestation were associated with high HOMA-IR in offspring. Consequently, the offspring of women with a combination of high folate and low B12 concentrations were found to be characterized by a significantly impaired insulin sensitivity status. These data support the idea that an inadequate or insufficient intake of micronutrients during intrauterine life leads to epigenetic changes in the fetus that may promote the development of chronic metabolic disease later in life [97]. Regarding minerals, the importance of an adequate supply of substances such as calcium, iodine, magnesium and zinc is well known. Calcium is a micronutrient that plays an important role in the bone mineralization of both the fetus and the mother, and whose action is closely related to vitamin D. Currently the recommended dose during pregnancy is 1 g/day and foods containing the highest amount of calcium are dairy products; consequently, in the case of vegetarian or vegan diets, a supplementation is recommended. Vitamin D deficiency is defined by values below 50 nmol/L according to the WHO and this is associated with several negative pregnancy and offspring outcomes, such as preeclampsia, GDM, preterm birth, LBW and SGA. Currently, the recommended daily dose is 600 IU/day. Iodine is an element present in very small amounts in our body which performs important functions for fetal growth and development. It represents the essential component of thyroid hormones, regulating carbohydrate, fat and protein metabolism, basal metabolism and development of the central nervous system. During the first weeks of gestation, the fetal thyroid hormone requirement is ensured by maternal synthesis until fetal synthesis is sufficient (about 17–19 weeks). It is very important to promptly recognize any iodine and thyroid hormone deficiency because even subclinical maternal hypothyroidism might have very serious consequences on the fetus, such as impaired neurocognitive development. The main food sources in which iodine is present are fish, meat, dairy products and iodized salt and the recommended daily dose during pregnancy is 200 μg. Magnesium is another important microelement that plays a role in glucose homeostasis. Indeed, it would seem to improve insulin sensitivity and reduce insulin resistance. An alteration in the magnesium-dependent channels may contribute to the onset of GDM. It is therefore important to ensure an adequate intake of magnesium, with the main food sources being legumes, nuts, seeds and green, leafy vegetables. In addition to magnesium, zinc would also appear to be involved in glucose metabolism and contribute to the control of GDM [97]. Thus, it is evident that micronutrients play a central role in metabolic control. Therefore, it is important to ensure an adequate intake of these substances in the diet, especially during pregnancy, in order to prevent the onset of metabolic diseases later in life and improve maternal and child health outcomes.

## 2. Conclusions

During the last few years, many observational studies and clinical trials on animals have been conducted which demonstrate the correlation between nutrition during various ages and the onset of glucose metabolism alterations. Therefore, the crucial importance of maternal nutrition during pregnancy and its effects on the metabolic programming of the fetus is now widely accepted. Particularly, an unbalanced maternal diet is known to correlate with both inadequate intrauterine growth and a higher risk of IR onset during postnatal life and, consequently, an increased risk of developing T2D and metabolic syndrome. In order to obtain a healthy and balanced maternal diet, an adequate supply of both macronutrients and micronutrients must be guaranteed. Particularly, it has been shown that a diet rich in foods containing low glycemic index carbohydrates, low saturated fat and an adequate protein content is correlated with better metabolic outcomes in the fetus. On the other hand, micronutrients in the maternal diet are also relevant components. Indeed, many studies have demonstrated that micronutrient deficiency, particularly vitamins B6, B9 and B12, affects the mechanisms of epigenetic regulation, promoting the development of metabolic alterations later in life. It is evident the great importance of adopting preventive strategies aimed at achieving a balanced diet in order to reduce the risk of metabolic diseases and, therefore, cardiovascular risk in adulthood. Furthermore, importance must be placed on an adequate intake of micronutrients, which has been shown to be decisive during intrauterine development.

## Figures and Tables

**Figure 1 nutrients-15-00087-f001:**
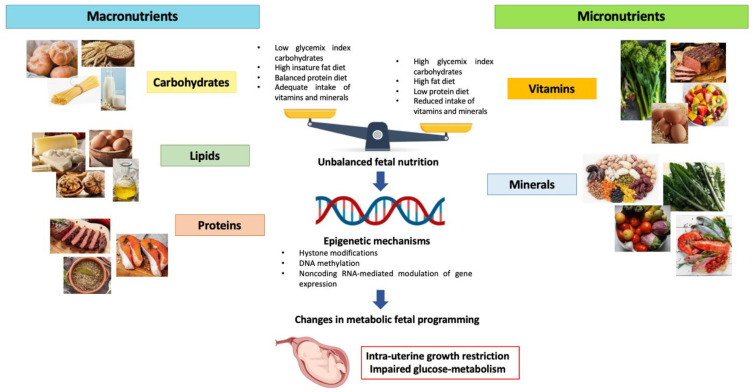
Effects of unbalanced maternal nutrition on intrauterine fetal growth and fetal metabolic programming.

**Table 1 nutrients-15-00087-t001:** Indirect methods of evaluating peripheral insulin sensitivity and their corresponding formulae.

INDIRECT METHODS	FORMULA
*HOMA-IR*	$insulin\;\left(\frac{\mu \mathrm{IU}}{L}\right)\times \frac{glucose\;\left(\mathrm{mmol}/L\right)}{22,5}$
*QUICKI*	$\frac{1}{\log insulin+\log glucose\;\left(in\frac{\mathrm{mg}}{\mathrm{dL}}\right)}$
*MCA*	$\exp\left(\begin{array}{c} \begin{array}{c} 2,63-0,28\;ln\\ \left(insulin\right)(\mu \mathrm{IU}/L) \end{array}\\ -0,31\ln\left(\begin{array}{c} triglycerides\\ \mathrm{mmol}/L \end{array}\right) \end{array}\right)$
*FIRI*	($fasting\;glucose\;(\mathrm{mmol}/L)\times \frac{FI\;(\mu \mathrm{IU}/L)}{25}$
*FPI*	*FPI*
*ISI*	$\begin{array}{c} {}[1,9/6\;x\;BW\;\left(\mathrm{kg}\right)\;x\;FPG\;(\mathrm{mmol}/L)+520\\ -1,9/18\;x\;BW\;x\;AUCFG\;(\mathrm{mmol}/h\cdot L)-UG(\mathrm{mmol}/1,8)]\\ /[AUCFI\;(\mathrm{pmol}/h\cdot L)\;x\;BW] \end{array}$
*WBISI*	$\frac{10,000}{\begin{array}{c} \sqrt{FPG\;in\;\mathrm{mg}/\mathrm{dL}\;x\;FPI\;in\;\mathrm{microU}/\mathrm{mL}}\\ \left(MG\;x\;MI\right) \end{array}}$

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
