# Peer review of "Role of Prenatal Nutrition in the Development of Insulin Resistance in Children"

_nutrients, 2022, doi:10.3390/nu15010087_

Round 1

Reviewer 1 Report

Overall comment: This is a succinct and well written narrative review covering many areas of interest in a complete way. I have only minor comments. 

Minor comment

Please double check the references. 

1.       (Page 1, Line 20): please spell out IR. It is the first time it appears in text.

2.       (Page 3, Line 69): Please spell out IUGR. It is the first time it appears in text.

3.       (Page 3, Line 74): Please spell out T2D. It is the first time it appears in text.

4.       (Page 5, Line 153-5): Please cite references for these three sentences.

Author Response

Overall comment: This is a succinct and well written narrative review covering many areas of interest in a complete way. I have only minor comments. 

We thank the reviewer for his/her helpful clarifications.

Minor comment

Please double check the references. 

  1. (Page 1, Line 20): please spell out IR. It is the first time it appears in text.
  2. (Page 3, Line 69): Please spell out IUGR. It is the first time it appears in text.
  3. (Page 3, Line 74): Please spell out T2D. It is the first time it appears in text.
  4. (Page 5, Line 153-5): Please cite references for these three sentences.

1) Page 1, Line 20: full spelling of IR has been added.

2) Page 3, Line 69: full spelling of IUGR has been added.

3) Page 3, Line 74: full spelling of T2D has been added.

4) Page 5, Line 153-5: references have been added.

Reviewer 2 Report

The authors reviewed the insulin resistance in children exposed to maternal malnutrion during early life. Although the changes of insulin resistance in children have been summarized, authors did not categolize or discuss which mechanisms of epigenetics is caused by, such as 1) DNA methylation, 2) Histone modification, 3) Non-coding RNA. 

Vitamin B12 and folate are well documented in epigenetics for the 1C metabolism, but correlation between other epigenetic changes and maternal malnutrition should also be elaborated.

It is desirable to devise ways to make it easier for readers to understand by using tables. The author's further systematic consideration is necessary, rather than listing the results reported so far. 

Major points

1)     Since the indices of insulin resistance are described in Section 1.2. and Table 1, it should be summarized or categolized in a table which indices were used/measured in the cited references.

2)    Authors should consider how long and which stage of pregnant of maternal malnutrition and overnutrition affect offspring's insulin resistance. In the case of animal experiments, it should be described and categorized at what age the offspring acquired insulin resistance.

3)    Glycemic index, high-fat, low protein, and micronutrients are explained by chapter, but minerals are missing in the manuscript. The important findings reported for the minerals, such as calcium and magnesium, should be cited and discussed.

4)    Authors should describe animal experiment data and human epidemiological data in separate chapter. Epidemic studies of insulin resistance in children with SGA also should be discussed in detail.

5)    Discussion should be given to the fact that male offspring are more susceptible than female. Estrogen may be one of the factors for the sex difference. In the later part of the low-protein chapter, thyroid hormone is also mentioned. It is difficult for readers to understand simply by listing hormones. It is necessary to explain their relevance in detail.

Minor points

1)      Although the full spellings of IR and T2D are shown in the abstract, the full spelling of the abbreviations is required at the first occurrence of the manuscript of introduction part.

2)      Line 183,184, 187: The notation of “beta-cell” and “β cells” is not unified.

Author Response

The authors reviewed the insulin resistance in children exposed to maternal malnutrion during early life. Although the changes of insulin resistance in children have been summarized, authors did not categolize or discuss which mechanisms of epigenetics is caused by, such as 1) DNA methylation, 2) Histone modification, 3) Non-coding RNA. 

Vitamin B12 and folate are well documented in epigenetics for the 1C metabolism, but correlation between other epigenetic changes and maternal malnutrition should also be elaborated.

It is desirable to devise ways to make it easier for readers to understand by using tables. The author's further systematic consideration is necessary, rather than listing the results reported so far. 

Major points

1)     Since the indices of insulin resistance are described in Section 1.2. and Table 1, it should be summarized or categolized in a table which indices were used/measured in the cited references.

1) We thank the referee for this important observation.

The aim of the manuscript is to describe the correlation between fetal exposure to intrauterine malnutrition/overnutrition and increased risk of the onset of insulin resistance. For greater completeness we have also discussed the methods of direct and indirect measurement of insulin resistance.

2)    Authors should consider how long and which stage of pregnant of maternal malnutrition and overnutrition affect offspring's insulin resistance. In the case of animal experiments, it should be described and categorized at what age the offspring acquired insulin resistance.

2) As has been suggested, we have better specified which stages of pregnancy are mostly correlated with an increased risk of the onset of insulin resistance in section 1.4. Regarding to animal studies, it is difficult to indicate a precise time of onset of insulin resistance in the offspring since it is influenced by several factors such as maternal diet during pregnancy and breastfeeding and the type of feeding after weaning. Intrauterine exposure to malnutrition lays the foundations in determining a genetic predisposition to develop insulin resistance, but it is not the only causal factor.

3)    Glycemic index, high-fat, low protein, and micronutrients are explained by chapter, but minerals are missing in the manuscript. The important findings reported for the minerals, such as calcium and magnesium, should be cited and discussed.

3) The suggested points have been added in section 1.8, concerning micronutrients, to have a more complete discussion. The most relevant minerals, which potentially play a role in the development of changes in metabolic programming of the fetus, were mentioned.

4)    Authors should describe animal experiment data and human epidemiological data in separate chapter. Epidemic studies of insulin resistance in children with SGA also should be discussed in detail.

4) We thank the reviewer for his/her comment.

The aim of the manuscript is to discuss about the close correlation between the intrauterine exposure of the fetus to unfavorable factors and the development of metabolic alterations during adulthood, and to explain the mechanisms underlying this correlation. Therefore, we thought it more appropriate to create sections describing the different maternal diets with their respective effects on fetal metabolism, focusing particularly on the role of micronutrients. In the manuscript, data extrapolated from animal studies or from epidemiological human studies have been cited, where necessary. Therefore, we believe that writing additional sections treating specifically about experimental animal studies and human epidemiological data would go beyond the aim of our manuscript. An insight into the correlation between SGA and insulin resistance has been added, as suggested, in section 1.4.

5)    Discussion should be given to the fact that male offspring are more susceptible than female. Estrogen may be one of the factors for the sex difference. In the later part of the low-protein chapter, thyroid hormone is also mentioned. It is difficult for readers to understand simply by listing hormones. It is necessary to explain their relevance in detail.

5) As has been suggested, we have expanded the comments relating to the differences between males and females in the development of insulin resistance, focusing mainly on the role played by estrogens. In addition, the correlation between maternal low-protein diet and thyroid dysfunction, mentioned in the previous version of the manuscript, was better explained. These changes were added in section 1.7, regarding the low protein maternal diet.

Minor points

1)      Although the full spellings of IR and T2D are shown in the abstract, the full spelling of the abbreviations is required at the first occurrence of the manuscript of introduction part.

1)  Full spellings of IR and T2D abbreviations has been added in the introduction part of manuscript at their first occurrence.

2)      Line 183,184, 187: The notation of “beta-cell” and “β cells” is not unified.

2) We have unified the notation of “beta cell” in the lines 183, 184, 187.

Round 2

Reviewer 2 Report

Need to double check typos and citation numbers.